# Enriched binocular experience followed by sleep optimally restores binocular visual cortical responses in a mouse model of amblyopia

Jessy D. Martinez [1], Marcus J. Donnelly[2], Donald S. Popke[2], Daniel Torres[1], Lydia G. Wilson[1], William P. Brancaleone[2], Sarah Sheskey[3], Cheng-mao Lin[3], Brittany C. Clawson[1], Sha Jiang[1] & Sara J. Aton [1✉]

Studies of primary visual cortex have furthered our understanding of amblyopia, long-lasting visual impairment caused by imbalanced input from the two eyes during childhood, which is commonly treated by patching the dominant eye. However, the relative impacts of monocular vs. binocular visual experiences on recovery from amblyopia are unclear. Moreover, while sleep promotes visual cortex plasticity following loss of input from one eye, its role in recovering binocular visual function is unknown. Using monocular deprivation in juvenile male mice to model amblyopia, we compared recovery of cortical neurons' visual responses after identical-duration, identical-quality binocular or monocular visual experiences. We demonstrate that binocular experience is quantitatively superior in restoring binocular responses in visual cortex neurons. However, this recovery was seen only in freely-sleeping mice; post-experience sleep deprivation prevented functional recovery. Thus, both binocular visual experience and subsequent sleep help to optimally renormalize bV1 responses in a mouse model of amblyopia.

[1] Department of Molecular, Cellular, and Developmental Biology, University of Michigan, Ann Arbor, MI, USA. [2] Undergraduate Program in Neuroscience, University of Michigan, Ann Arbor, MI, USA. [3] Department of Ophthalmology and Visual Sciences, University of Michigan Medical School, Ann Arbor, MI, USA. ✉email: saton@umich.edu

During early postnatal development, both experience-driven synaptic plasticity and sleep impact lifelong sensory and behavioral functions[1–3]. For example, MD (occlusion of one of the two eyes) early in life shifts responsiveness of bV1 neurons to favor the dominant eye[4,5]—a process known as ocular dominance plasticity (ODP). ODP results from depression of deprived eye (DE) responses, followed by potentiation of spared eye (SE) responses, in bV1 neurons[6,7]. Sleep plays an essential role in promoting ODP during the critical period, promoting both synaptic strengthening and weakening in V1 in the hours following monocular visual experience[8–11].

ODP is a model for the neural mechanisms underlying amblyopia, a visual disorder caused by imbalanced input to bV1 from the two eyes in early childhood, leading to long-term disruption of binocular vision and poor visual acuity[12–15]. Dominant eye patching is the standard clinical intervention to promote recovery in amblyopia. This strategy was established based on studies carried out in both cats and monkeys, in which occlusion of the previously-dominant eye (reverse occlusion; RO) was sufficient to drive a somewhat greater recovery of DE responses in bV1 than reopening the DE alone[16,17]. Critically, however, neither RO nor simply reopening the DE restored binocularity of bV1 neurons' visual responses[17], and numerous studies have found long-lasting visual deficits after RO, despite recovery of DE responses[18–20]. More recently, studies in developing cats and rodents have found that under certain conditions, binocular vision can serve to restore binocularity of responses in bV1[21–25]. Across species and developmental stages, binocular presentation of high-contrast stimuli such as gratings that synchronously activate left and right eye pathways (leading to coincident activation of bV1 neurons) seems to be an optimal driver of recovery[24,26]. Thus, intensive binocular experience—aimed at promoting cooperative input from the two eyes to bV1—has recently been explored as a therapeutic strategy for recovery in amblyopic patients[27–32]. It remains unclear whether binocular or monocular interventions are superior at restoring vision to amblyopic children—with randomized clinical trials using dichoptic iPad games to provide binocular stimulation yielding conflicting results[13,32,33]. It thus remains unclear: (1) whether differences in recovery are apparent when the duration and quality (e.g., with identical contrast, spatial frequency, and temporal features) of visual stimuli are carefully controlled, and (2) what changes to the bV1 network (e.g. in visual responses of excitatory vs. inhibitory neurons) mediate these differences.

Sleep can benefit processes relying on synaptic plasticity, including ODP in bV1[8–11,34–36]. In cat V1, initial shifts in ocular dominance following a brief period of MD are augmented by a few hours of subsequent sleep[10] and are disrupted by sleep deprivation (SD)[11]. This suggests that sleep immediately following either monocular (RO) or binocular (BR) recovery experiences could also promote recovery of bV1 function after a period of MD. However, in a single study in critical period cats, a period of sleep following a brief interval of post-MD RO actually impaired (rather than enhanced) recovery of normal V1 ocular dominance[37]. Thus, the function of appropriately-timed sleep in promoting (or disrupting) visual cortical responses in amblyopia—particularly after binocular visual recovery—remains to be determined.

To address these questions, we first directly compared how multi-day, post-MD BR and RO—of identical duration and visual stimulus content—affect recovery of function in mouse binocular V1 (bV1). Using single-neuron recordings, we find that bV1 ocular dominance shifts caused by 5-day MD are completely reversed by a period of visually-enriched BR experience (in a scenario where high-contrast, dynamic stimuli are delivered to the two eyes simultaneously), but are only partially reversed by RO of identical duration and quality. These differential effects were observed in both regular spiking (RS) neurons and fast spiking (FS; putative

parvalbumin-expressing [PV +]) interneurons. BR, but not RO, reversed MD-induced depression of DE-driven firing rate responses in both RS neurons and FS interneurons, and increases in SE-driven responses in both populations. Recovery of bV1 visual function was confirmed by quantifying DE-driven cFos expression, which was reduced in layers 2/3 after MD (across the population as a whole, and among PV + interneurons), and recovered to control levels after BR, but not RO. Critically, BR-driven recovery of ocular dominance, bV1 visual response changes, and DE-driven cFos expression were all disrupted by SD in the hours immediately following periods of visual experience. Together, these results suggest that optimal recovery of bV1 function after a period of MD is promoted by enriched binocular visual experience and subsequent, undisturbed sleep. These data add to a growing body of literature that suggest potential alternative strategies for treatment of amblyopia, that may improve upon the current gold standard for clinical care (dominant eye patching, with no emphasis on relative sleep timing).

## Results

### Binocular recovery (BR) causes more complete reversal of MD-induced bV1 ocular dominance shifts than identical-duration reverse occlusion (RO).
We first directly compared the degree of bV1 response recovery induced by multi-day BR and RO in bV1 neurons following a 5-day period of MD (Fig. 1a). The duration and timing of MD (P28-33; during the peak of the critical period for ODP) was chosen with the aim of inducing a robust ocular dominance shift, with changes to both DE and SE responses in bV1[6]. To ensure comparable quality and duration of visual experience between BR and RO recovery groups, and to optimize potential for recovery of binocular responses, from P33-38, these mice were placed for 4 h/day (starting at lights-on) in a square chamber surrounded by four LED monitors presenting high-contrast, phase-reversing gratings (8 orientations, 0.05 cycles/deg, reversing at 1 Hz) in an interleaved manner. This type of visual stimulus was selected to mimic visual stimuli used experimentally to promote recovery in amblyopia patients[30,32,33,38,39], and is similar to stimuli that promote optimal recovery from MD in adult mice[24]. With binocular presentation, this visual enrichment would optimize for synchronous co-activation of inputs to V1 representing the two eyes, which is thought to be an important feature of visual response recovery. During this period of visual enrichment, mice had access to a running wheel, manipulanda, and treats in order to increase wake time, promote more consistent visual stimulation, and drive maximum recovery[24]. After the 5-day recovery period, we compared bV1 neurons' visual responses for stimuli presented to either the right or left eyes, for the hemisphere contralateral to the original DE (Fig. 1b).

Consistent with previous reports, 5-day MD induced a large ocular dominance shift in favor of the SE compared to normally-reared (NR) control mice with binocular vision from birth (Fig. 1c–e). Five days of BR visual experience returned bV1 ocular dominance to a distribution similar to age-matched NR mice, completely reversing the effects of MD. After BR, ocular dominance index distributions (Fig. 1d) and contralateral bias indices for each mouse (Fig. 1e) matched those of NR mice, showing a preference for the DE (contralateral) eye. In contrast, ocular dominance distributions following 5-day RO visual experience were intermediate between MD mice and age-matched NR mice (Fig. 1c-e), suggesting only partial recovery. Visual responsiveness among neurons within bV1 was similar between the groups. The proportion of visually responsive recorded neurons (i.e., those with higher firing rate responses to grating stimuli than to blank screen presentation) was 84.4% (222/263), 85.6% (238/278), 85.8% (230/268), and 82.8% (217/262) for NR, MD, BR, and RO groups, respectively.

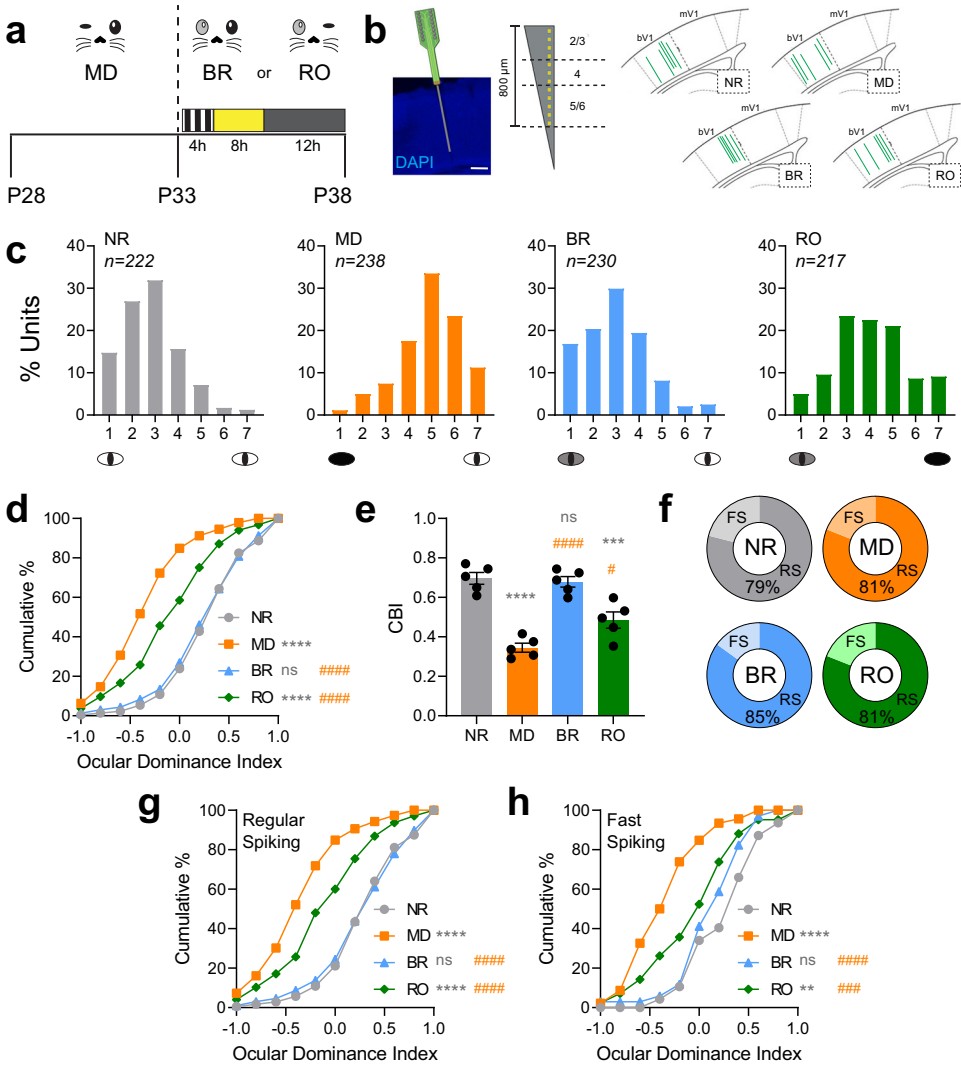

**Fig. 1 BR is more effective than RO at reversing MD-induced ocular dominance shifts. a** Experimental design. Mice underwent 5-day MD from P28-P33. MD mice were recorded at P33. Two recovery groups with either binocular recovery (BR) or reverse occlusion (RO) visual experience from P33-38 had daily 4-h periods of visual enrichment starting at lights on and were recorded at P38. Normally-reared (NR) mice were recorded at P38 without prior manipulation of vision. **b** Representative image of electrode probe placement in binocular primary visual cortex (bV1) coronal section stained with DAPI and enlarged view of electrode contacts, which spanned the layers of bV1 (scale bar = 200 μm). Schematic of bV1 coordinates in coronal sections where green lines represent probe placements in bV1 for all groups. **c** Ocular dominance histograms from bV1 neurons recorded contralateral to the original DE for all four groups, using a 7-point scale (1 = neurons driven exclusively by contralateral eye; 7 = neurons driven exclusively by ipsilateral eye, 4 = neurons with binocular responses) n = 5 mice/group. **d** Cumulative distribution of ocular dominance indices for all neurons recorded in each group. **e** Contralateral bias indices for mice in each treatment group. One-way ANOVA: $F_{(3, 16)} = 29.34$, $p < 0.0001$. Error bars indicate mean ± SEM. **f** The proportion of recorded neurons classified as regular spiking (RS) neurons and fast-spiking (FS) interneurons in each treatment group. RS neurons: NR (n = 175); MD (n = 192); BR (n = 196); RO (n = 175). FS interneurons: NR (n = 47); MD (n = 46); BR (n = 34); RO (n = 42). **g, h** Ocular dominance index cumulative distributions for RS neurons (**g**) and FS interneurons (**h**). Ocular dominance index values for both populations were significantly shifted in favor of the SE after MD, were comparable to those of NR mice after BR, and were intermediate—between NR and MD values—after RO. **, ***, and **** (gray) indicate $p < 0.05$, $p < 0.01$ and $p < 0.0001$, K-S test vs. NR (**d, g, h**) or Tukey's post hoc test vs. NR (**e**); #, ### and #### (orange) indicate $p < 0.05$, $p < 0.001$ and $p < 0.0001$, K-S test vs MD (**d, g, h**) or Tukey's post hoc test vs MD (**e**); ns indicates not significant.

MD is known to effect a change in the balance of activity between principal (RS; mainly glutamatergic) neurons and FS (mainly PV +, GABAergic) interneurons[9,40,41]. In our extracellular recordings, FS interneurons (identifiable based on firing rate and distinctive spike waveform features—i.e., narrower spike half-width; Supplementary Fig. 1[9,42]) represented roughly 15–20% of all stably-recorded neurons (i.e., those with spiking present across the entire visual response testing period), across all treatment conditions (Fig. 1f). We found that relative to neurons recorded from NR mice, MD led to similar ocular dominance shifts toward the SE in both RS neurons and FS interneurons

(Fig. 1g and 1h, respectively). These MD-induced changes were completely reversed in both RS and FS populations in BR mice, but were only partially reversed in RO mice (Fig. 1f-g). We conclude that near the closure of the critical period for ODP, 5-day BR is quantitatively superior to 5-day RO at reversing effects of MD.

**BR and RO differentially restore bV1 RS neuron and FS interneuron firing rate responses after MD.** MD leads to sequential changes in V1 neurons' maximal responses to DE

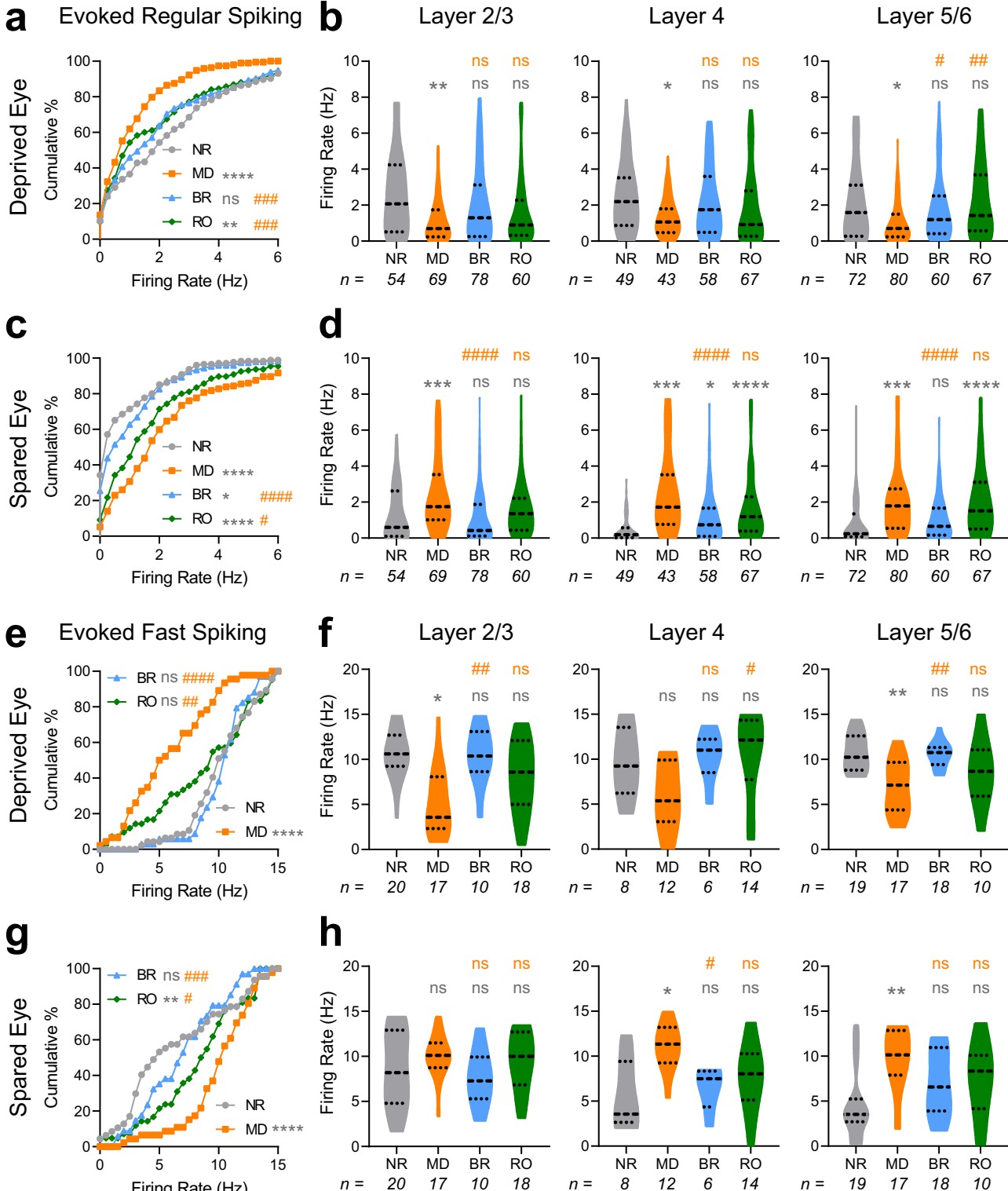

and SE stimulation (which are depressed and potentiated, respectively)[6,7,9,10,43]. We next investigated which of these changes could be reversed in bV1 neurons as a function of post-MD BR or RO. To better characterize microcircuit-level changes due to MD, we examined how DE and SE visual response recovery varied between RS neuron and FS interneuron populations, and in different layers of bV1. DE responses were significantly depressed after 5-day MD as previously reported[6,7]; these changes were seen across cortical layers, in both RS neurons

(Fig. 2a-b) and FS interneurons (Fig. 2e, f). In both populations, DE response depression was most pronounced in the extragranular layers. Both BR and RO both largely reversed DE response depression in RS neurons, although modest differences remained after RO (Fig. 2a); recovery appeared most complete in RS neurons in layers 5/6 (Fig. 2b). DE response depression in FS interneurons was fully reversed by 5-day BR (Fig. 2e), with the most dramatic changes occurring in the extragranular layers (Fig. 2f). In comparison, response depression reversal was more

**Fig. 2 BR and RO differentially reverse MD-induced changes in DE and SE firing rate responses among RS neurons and FS interneurons.**
**a**, **c** Cumulative distributions of preferred-stimulus (i.e. maximal) DE (**a**) and SE (**c**) visually-evoked firing rate responses for bV1 RS neurons. DE responses were significantly depressed after 5-day MD; this was reversed fully after BR and partially after RO. SE responses in RS neurons showed post-MD potentiation, which was maintained after RO, but largely reversed by BR. **b**, **d** Violin plots of RS neurons' DE (**b**) and SE (**d**) visually-evoked responses recorded from neurons in bV1 layers 2/3, 4, or 5/6. Kruskal–Wallis test, $p = 0.009$, $p = 0.035$, $p = 0.002$ for layers 2/3, 4, and 5/6 in the DE, respectively. Kruskal–Wallis test, $p < 0.0001$, $p < 0.0001$, $p < 0.0001$ for layers 2/3, 4, and 5/6 in the SE, respectively. **e**, **g** Cumulative distributions of maximal DE (**e**) and SE (**g**) visually-evoked firing rate responses for FS interneurons. DE and SE responses were depressed and potentiated, respectively, after MD. These response changes were partially reversed by RO, and fully reversed by BR. **g**, **h** Violin plots of FS interneurons' DE (**f**) and SE (**h**) visually-evoked responses recorded from neurons in each bV1 layer. Kruskal–Wallis test, $p = 0.002$, $p = 0.021$, $p = 0.0004$ for layers 2/3, 4, and 5/6 in the DE, respectively. Kruskal–Wallis test, $p = 0.29$, $p = 0.005$, $p = 0.006$ for layers 2/3, 4, and 5/6 in the SE, respectively. *, **, *** and **** (gray) indicate $p < 0.05$, $p < 0.01$, $p < 0.001$, and $p < 0.0001$, K-S test vs. NR (**a**, **c**, **e**, and **g**) or Dunn's *post hoc* test (**b**, **d**, **f**, and **h**); #, ##, ### and #### (orange) indicate $p < 0.05$, $p < 0.01$, $p < 0.001$, and $p < 0.0001$, K-S test vs MD (**a**, **c**, **e**, and **g**) or Dunn's *post hoc* test (**b**, **d**, **f**, and **h**); ns indicates not significant. Dashed lines in violin plots (**b**, **d**, **f**, and **h**) represent the 25%, median, and 75% quartiles. Sample sizes per group of units found in figure.

modest (though still significant) after RO (Fig. 2e), with the largest changes occurring in layer 4 FS interneurons (Fig. 2f).

MD strongly potentiated responses to SE stimulation, across both bV1 neuron populations, and across cortical layers (Fig. 2c, d, g, h). BR and RO had differential effects with respect to reversing MD-potentiated responses. For both RS neurons and FS interneurons (Fig. 2c, g), potentiation of SE responses was almost completely reversed by BR. In contrast, in both neuron populations, RO led to only partial reversal of MD-induced SE response potentiation (Fig. 2c, g). After BR, reversal of SE response potentiation was present in RS neurons across bV1 layers. In contrast, after RO, SE responses remained significantly potentiated in layer 4 and layers 5/6 (Fig. 2d). Among FS interneurons, BR tended to reverse SE response potentiation more completely than RO across all layers of bV1, with the most complete reversal (leading to significant differences from MD alone) seen in layer 4 (Fig. 2h). Together, these data suggest that 5-day BR is superior to RO with respect to reversing both synaptic depression and synaptic potentiation in bV1 caused by prior MD.

MD-driven changes in bV1 neurons' visually-evoked firing responses reflect a combination of Hebbian and homeostatic plasticity mechanisms[44–48]. While Hebbian plasticity mechanisms (i.e., LTP and LTD of glutamatergic synapses) have the potential to alter maximal firing rate responses to visual stimulation, homeostatic plasticity could also affect the spontaneous firing rate of bV1 neurons. To assess how MD and recovery experiences affect the overall firing of bV1 neurons, we also compared mean (across all visual stimulus presentations) and spontaneous (during blank screen presentation) firing rates in bV1 after MD and recovery experience (Supplementary Fig. 2). Among RS neurons, mean (Supplementary Fig. 2a) and spontaneous (Supplementary Fig. 2c) responses for the DE were unaffected or augmented, respectively—effects which differed from the depression of DE maximal firing rate responses observed after MD (Fig. 2a). This difference suggests that while peak DE visual responses are depressed after MD (via a Hebbian mechanism), this effect may be partially offset by homeostatic firing enhancement. While RO had no further effect on DE mean or spontaneous firing rates, BR enhanced overall firing in RS neurons (i.e., beyond levels seen after MD). SE mean and spontaneous firing in RS neurons were enhanced by MD; these effects were not altered by RO and were only partially reversed after BR. Among bV1 FS interneurons, DE mean and spontaneous firing rate changes after MD, RO, and BR (Supplementary Fig. 2b, d) followed the same pattern as changes in maximal firing rates (Fig. 2e)—with suppression after MD which were reversed after both recovery experiences. SE mean and spontaneous firing was modestly, but significantly, enhanced in FS neurons after MD; this enhancement remained after RO, but not BR. Taken together, these data suggest that many, but not all, of the response changes observed in bV1 neurons' preferred-orientation responses

after visual manipulations are also observed in their mean and spontaneous firing.

**BR, but not RO, fully restores DE-driven cFos expression in bV1 layers 2/3.** To further characterize how MD, BR, and RO affect visual responses throughout bV1, we used immunohistochemistry to quantify DE-driven cFos expression in PV + interneurons and non-PV + neurons. Mice were treated as shown in Fig. 1, after which they were returned to the visual enrichment arena for 30 min of visual stimulation of the DE only, then were perfused 90 min later. Visually-driven expression of cFos and PV expression were quantified across the layers of bV1 contralateral to the DE. Consistent with previous reports[49,50], DE-driven cFos expression was significantly reduced across bV1 after MD (Fig. 3a). Both total density of cFos+ neurons and the density of cFos+ PV + interneurons decreased after MD. BR reversed these changes, restoring DE-driven cFos expression to levels seen in NR control mice (Fig. 3a, c). In contrast, and consistent with data shown in Figs. 2a and 2d, both total DE-driven cFos expression and density of cFos+ PV + interneurons remained significantly reduced after RO. Quantification of cFos and PV by layer showed that the largest differential effects of visual experience were seen in layers 2/3. Following MD, DE-driven cFos expression was reduced across all layers (Fig. 3d), and cFos+ PV + interneuron density was dramatically reduced in layers 2/3 (and to a lesser extent, layer 4) (Fig. 3f). RO restored DE-driven cFos expression in layer 4 and layers 5/6, but not layers 2/3 (Fig. 3d, f). After BR, total and PV + interneuron cFos expression was renormalized across all layers, including layer 2/3, where cFos+ neuron density was restored to levels seen in NR mice. As an additional control, to verify that changes in expression were driven by alterations of visual input, the same analysis was applied to adjacent segments of primary auditory cortex (A1) within the same immunolabeled brain sections used for bV1 measurements. As shown in Supplementary Fig. 3, no differences in cFos or PV expression were observed between the four experimental groups. Together, these data suggest that activity-driven plasticity in layer 2/3, especially in layer 2/3 PV + interneurons, differs dramatically during monocular vs. binocular recovery from MD.

**BR is modestly superior to RO in preventing visual acuity losses after MD.** A common feature of amblyopia is long-lasting changes in visual acuity[51] for both deprived and spared eyes. We tested whether differences in V1 after BR vs. RO are associated with differences in visual function, using DE- and SE-driven optokinetic responses (Supplementary Fig. 4a) as a metric of visual acuity for the two eyes. Acuity measured for the two eyes in NR mice was similar to that reported previously for mice tested using optokinetic responses during the critical period[52,53], and

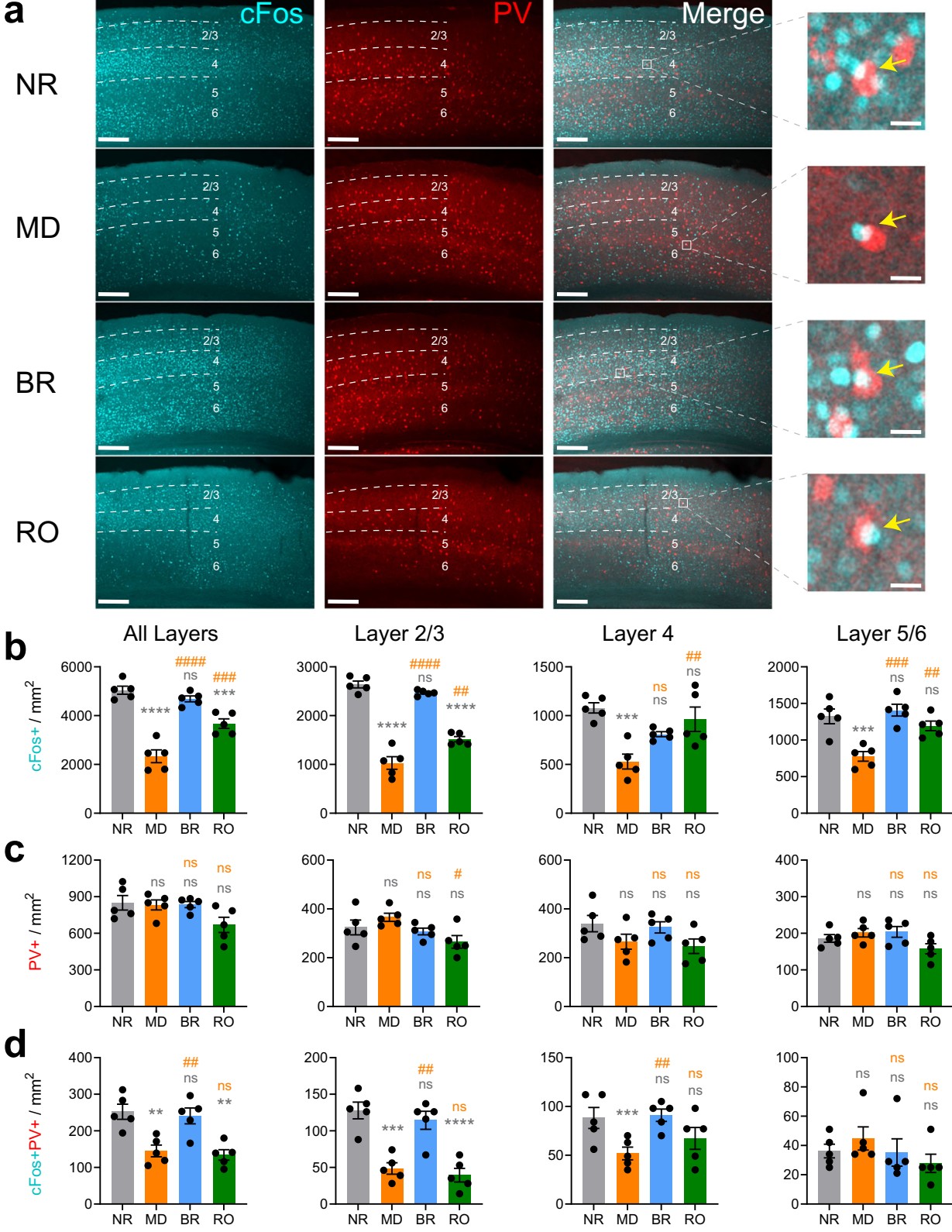

slightly lower than that of adult mice tested using either the visual water maze[54] or optokinetic responses[52]. Consistent with previous findings[53], 5-day MD reduced DE acuity, and had no significant effect on SE acuity, relative to NR mice (Supplementary Fig. 4b). 5-day BR led to a partial recovery of DE acuity, but consistent with previous findings in rats[22], slightly reduced SE

acuity. Finally, and consistent with prior work in cats[20] 5-day RO led to significantly reduced visual acuity in both eyes. Taken together, these data suggest that differential response properties in V1 for BR vs. RO mice are associated with subtly different outcomes for visual acuity, with BR mice showing less disruption of vision than RO mice.

**Fig. 3 DE-driven cFos expression is reduced after MD and restored after BR, but not RO. a** Representative images of bV1 cFos (cyan), parvalbumin (PV) [red], and overlap across treatment groups following DE stimulation. Mice ($n = 5$/treatment group) received DE-only visual stimulation for 30 min, then were returned to their home cages for 90 min prior to perfusion. Dashed lines represent cortical layer distribution used in cell counting analysis. Scale bar = 100 µm; 20 µm (magnification inset). **b** DE-driven cFos+ neuron density was decreased in bV1 after MD. cFos expression was fully rescued after BR and partially rescued after RO. One-way ANOVA: $F (3, 16) = 39.65$, $p < 0.0001$. cFos+ neuron density in bV1 layers 2/3, 4, and 5/6. One-way ANOVA for layers 2/3, 4, or 5/6, respectively: $F (3, 16) = 95.41$, $p < 0.0001$, $F (3, 16) = 9.093$, $p = 0.001$, and $F (3, 16) = 12.35$, $p = 0.0002$. **c** Density of PV + bV1 interneurons was similar between groups. One-way ANOVA: $F (3, 16) = 2.99$, $p = 0.062$. PV + interneuron density in bV1 layers 2/3, 4, and 5/6. One-way ANOVA for layers 2/3, 4, or 5/6, respectively: $F (3, 16) = 3.40$, $p = 0.044$, ns, and ns. **d** cFos+ PV + interneuron density decreased with MD and recovered with BR, but not RO. One-way ANOVA: $F (3, 16) = 11.40$, $p = 0.0003$. cFos+PV + interneuron density in bV1 layers 2/3, 4, and 5/6. One-way ANOVA for layers 2/3, 4, or 5/6, respectively: $F (3, 16) = 18.88$, $p < 0.0001$, $F (3, 16) = 4.25$, $p = 0.022$, and ns. \*\*, \*\*\*, and \*\*\*\* (gray) indicate $p < 0.01$, $p < 0.001$, and $p < 0.0001$, Tukey test vs. NR; #, ##, ### and #### (orange) indicate $p < 0.05$, $p < 0.01$, $p < 0.001$, and $p < 0.0001$, Tukey test vs MD; ns indicates not significant. Error bars indicate mean ± SEM.

**Sleep in the hours following BR visual experience is necessary for ocular dominance recovery.** Initial shifts in ocular dominance in favor of the SE are promoted by periods of sleep following monocular visual experience[9–11,55]. However, it is unclear whether, or how, sleep contributes to bV1 functional recovery after MD. Because 5-day BR (with 4 h of binocular visual enrichment per day) was effective at reversing many of the effects of prior MD, we tested whether post-visual enrichment sleep plays an essential role in this recovery. Mice underwent the same 5-day MD and 5-day BR periods shown in Fig. 1. Following each daily visual enrichment period, mice were returned to their home cage, and over the next 4 h were either sleep deprived (SD) under dim red light (to prevent additional visual input to V1) or allowed *ad lib* sleep (BR + SD and BR + Sleep, respectively; Fig. 4a-b). BR + Sleep mice spent (on average) just over 70% of the first 4 h post-enrichment period (corresponding to SD in BR + SD mice) in a behavioral sleep posture (crouched, immobile, nested and with closed eyes) (Fig. 4c). We then compared bV1 neurons' visual responses for stimuli presented to either the right or left eyes, for the hemisphere contralateral to the original DE, between BR + Sleep and BR + SD mice (Fig. 4d). In contrast to prior reports on the effects of SD following RO in critical period cats[37], we found that SD in the hours following daily BR visual experience reduced post-MD recovery of ocular dominance in favor of the original DE (Fig. 4e). Ocular dominance index and contralateral bias index values for bV1 neurons recorded from BR + SD mice were significantly reduced compared to those of BR + Sleep mice, indicating reduced DE preference similar to that seen after MD alone (Fig. 4f-g). RS neurons and FS interneurons were discriminable based on firing rate and spike half-width (Supplementary Fig. 5). The effects of SD on ocular dominance recovery across BR were present in both RS neurons and FS interneurons in bV1 (Fig. 4h-j). The proportion of visually responsive neurons recorded in BR + Sleep and BR + SD mice was also similar (88.3% and 84.5%, respectively). Thus, in the context of BR-mediated recovery from MD, post-experience sleep plays an essential role in recovery of ocular dominance in bV1.

**BR-mediated renormalization of DE and SE responses are reversed by sleep loss.** To determine how SD affects visual responsiveness in DE and SE pathways, we assessed how maximal visually-evoked firing rates (at each neuron's preferred stimulus orientation) were affected by post-BR sleep vs. SD. In both RS neurons and FS interneurons, post-BR SD led to a significant reduction of DE firing rate responses compared with those recorded from freely-sleeping mice (Fig. 5a-b, e-f). When DE responses were compared across bV1 as a whole, those recorded from BR + SD mice were significantly lower than those recorded from BR + Sleep mice, similar to those recorded from mice after MD alone. Among RS neurons, we found that this effect was most pronounced in layers 2/3, where DE-driven firing rates in BR +

SD mice were similar to those recorded from MD mice (Fig. 5b). Among FS interneurons, SD effects were most pronounced in layer 4, where depressed DE responses were similar to those of MD-only mice (Fig. 5f).

Across bV1 as a whole, RS neurons' SE responses were not significantly different between BR + Sleep and BR + SD mice (Fig. 5c). SE responses were significantly elevated in RS neurons recorded in layer 4 and layers 5/6 from BR + SD mice, where median response rates were similar to those recorded in MD-only mice (Fig. 5d). Across the bV1 FS interneuron population, SD interfered with BR-driven normalization of SE responses, which remained elevated, similar to those recorded from mice following MD alone (Fig. 5g). FS interneurons' SE responses in BR + SD mice were significantly elevated relative to BR + Sleep mice in layers 5/6, with median firing rate responses similar to those seen in MD mice (Fig. 5h). Together, these data suggest that eye-specific response renormalization due to BR in both RS neurons and FS interneurons is suppressed by post-BR SD.

As with MD, BR, and RO groups (Supplementary Fig. 2), we also compared mean (i.e., across all visual stimulus orientations) and spontaneous (blank screen) firing between BR + Sleep and BR + SD groups (Supplementary Fig. 6), to identify sleep-dependent changes that might be mediated by homeostatic, rather than Hebbian, mechanisms. As shown in Supplementary Fig. 6a and c, both DE and SE mean and spontaneous firing followed a similar pattern to maximal visually-evoked firing (Fig. 5a, c). Because both SE mean and spontaneous firing differed between the two groups, with SE firing being reduced overall in BR + Sleep mice (and this was most pronounce for spontaneous firing (Supplementary Fig. 6c)), one possibility is that this difference is driven by homeostatic synaptic downscaling occurring in a sleep-dependent manner in bV1[56]. FS interneurons' mean and spontaneous firing rates generally did not differ significantly between BR + Sleep and BR + SD mice (Supplementary Fig. 6b–d), although SE mean firing rates were lower in BR + Sleep mice, similar to maximal visually-evoked firing (Fig. 5G).

To further characterize layer- and cell type-specific changes in visual responses after post-BR sleep vs. SD, we quantified DE-driven cFos expression in bV1 of BR + Sleep and BR + SD mice, using the same DE visual stimulation strategy described for Fig. 3. Across bV1 as a whole, overall DE-driven cFos expression was significantly reduced in BR + SD mice compared to BR + Sleep mice (Fig. 6a). This reduction was most dramatic in layers 2/3 and 5/6 (Fig. 6d), where cFos levels in BR + SD mice were intermediate between those of BR + Sleep and MD-only mice. The density of cFos+ PV + interneurons was likewise significantly decreased after DE stimulation in BR + SD mice (Fig. 6c), with dramatic reductions in layers 2/3 and 4 (Fig. 6f). As an additional control, to verify that changes in expression were driven by alterations of visual input, the same analysis was applied to adjacent segments of primary auditory cortex (A1) within the same immunolabeled brain sections used

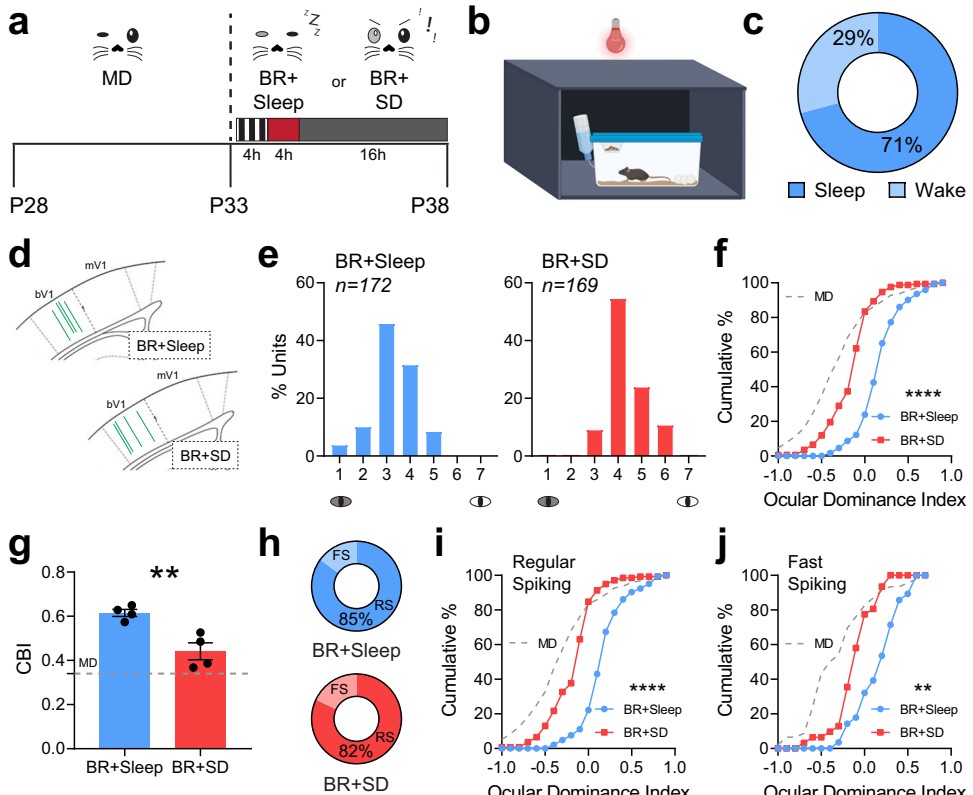

**Fig. 4 Sleep loss following BR visual experience prevents ocular dominance shifts. a** Experimental design. Mice underwent 5-day MD and 5-day BR; each day after 4-hr BR, BR + Sleep mice were returned to their home cage and allowed *ad lib* sleep under dim red light, BR mice underwent 4 h of sleep deprivation (BR + SD) through gentle handling under dim red light. **b** Schematic of experimental setup for animal observation under dim red light. **c** On average, BR + Sleep mice spent 71% of the 4 h period following visual enrichment (n = 4) in sleep, based on visual confirmation of immobility, stereotyped (crouched) sleep postures, nesting, and closed eyes, consistent with prior studies[83,102–104]. **d** Schematic of bV1 coordinates in coronal sections. Green lines indicate probe placements in bV1 for BR + Sleep and BR + SD groups. **e** Ocular dominance histograms for bV1 neurons recorded from BR + Sleep and BR + SD groups (4 mice/group). **f** Cumulative distribution of ocular dominance index values for bV1 neurons recorded from BR + Sleep and BR + SD mice. Values from neurons recorded in MD-only mice from Fig. 1 are shown (dashed gray lines) for comparison. **g** Contralateral bias index values were reduced for bV1 neurons recorded from BR + SD mice. Unpaired t-test: p = 0.0059. Error bars indicate mean ± SEM. **h** Proportion of recorded neurons identified as RS neurons or FS interneurons for the two groups. RS neurons: BR + Sleep (n = 144); BR + SD (n = 138). FS interneurons: BR + Sleep (n = 28); BR + SD (n = 31). **i, j** Ocular dominance index values for recorded RS neurons (**i**) and FS interneurons (**j**) were reduced in BR + SD mice. ** and **** indicate p < 0.01 and p < 0.0001, K-S test.

for bV1 measurements. As shown in Supplementary Fig. 7, no differences in cFos or PV expression were observed between BR + Sleep and BR + SD groups. Taken together, our data suggest that most of the changes to DE and SE responses initiated in bV1 by MD are sustained when BR is followed by SD. Conversely, BR-mediated recovery of binocular function in bV1 RS neurons and FS interneurons relies on post-BR sleep.

## Discussion

In this study, we compared how different recovery experiences and subsequent sleep affect recovery of visual cortical responses following MD. We first compared the effects of equal-duration, qualitatively-similar binocular vs. monocular visual experience on recovery of bV1 responses following MD, using single-unit electrophysiology and immunohistochemistry. It is important to note that our BR intervention did not involve simply re-opening the original DE—an intervention used in early studies using primates and cats as an amblyopia model[16,17]. Rather, we attempted to create an experimental scenario for BR where cooperative input from the two eyes might be expected to reach bV1 simultaneously, and that mice would be encouraged to use the eyes together. We chose to optimize visual stimulation during recovery experiences with high-contrast gratings, which have been shown to optimize recovery of

visual function in adult mice after MD[24,26], as well as providing coincident activation of the two eyes during binocular viewing. Moreover, we place BR and RO mice in a behavioral scenario where: (1) the stimuli were novel, and (2) as BR mice moved through the environment, binocular disparity cues (i.e., spatial frequency of stimuli presented to the eyes) would be constantly changing. Critically, a side-by-side comparison of equal-duration BR and RO clearly showed that bV1 ocular dominance shifts in favor of the SE are reversed after 5-day BR, but not 5-day RO (Fig. 1). This reversal is present in both RS neurons and FS interneurons, and is associated with reversal of both MD-driven DE response depression and SE response potentiation (Fig. 2, Supplementary Fig. 2). Insofar as MD serves as a model for amblyopia caused by disruption of vision in one of the two eyes during childhood, these data add to a body of growing evidence that suggests that enriched binocular visual experience may offer advantages over and above the standard of care for amblyopia[21–25]. We also characterized the effects of post-experience sleep and sleep loss on recovery processes. When daily BR experience is followed by SD, recovery of normal binocular vision in bV1 is nearly completely blocked (Figs. 4–6). This suggests that the relative timing of sleep relative to recovery experience is potentially a critical—but overlooked—consideration for the treatment of amblyopia.

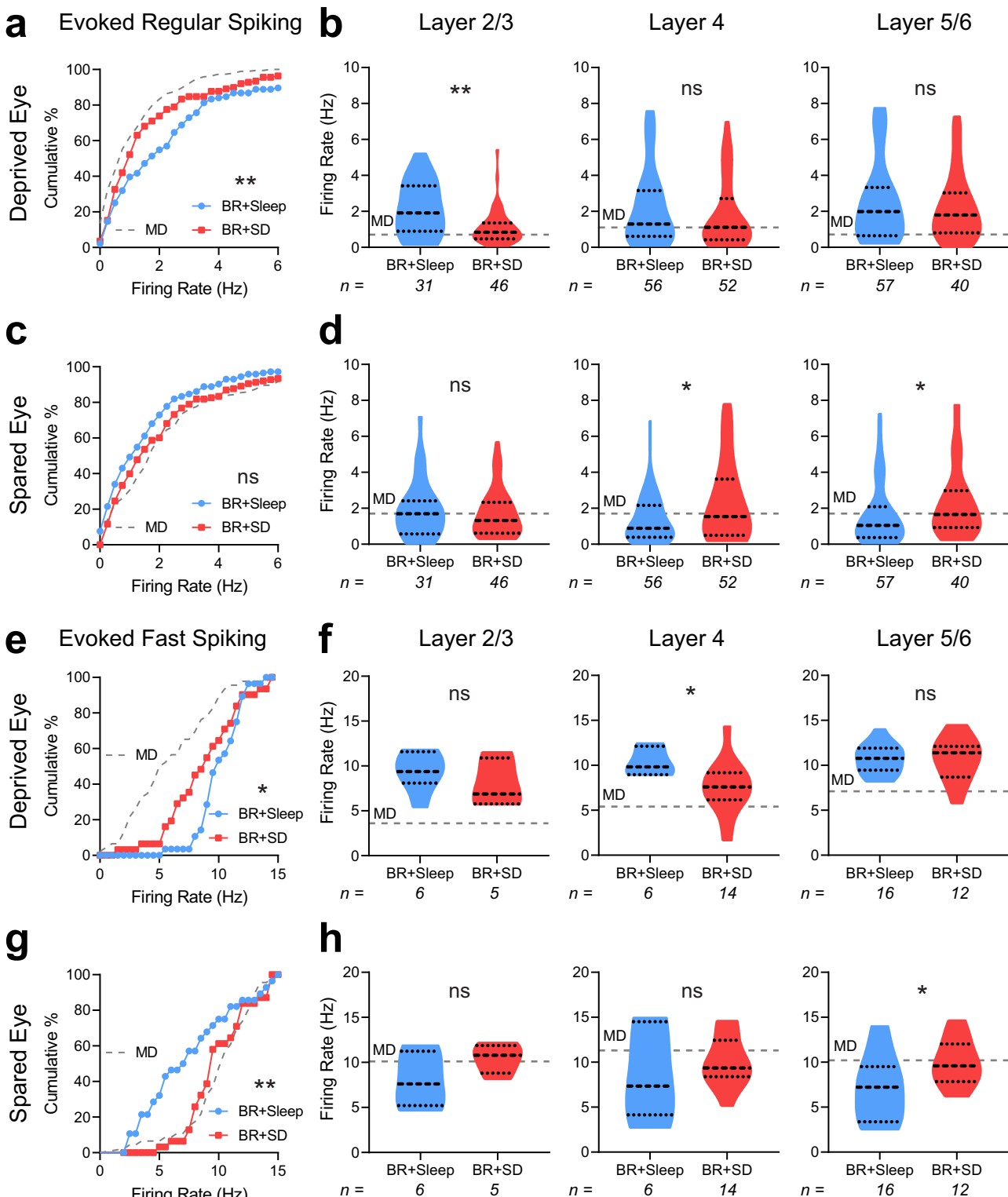

How do BR and RO differ in their effects in bV1? Here we find that 5-day MD causes significant DE response depression among both RS neurons and FS interneurons in bV1—with the most dramatic depression observed among FS interneurons in layers 2/3. These findings are consistent with results of longitudinal calcium imaging studies in mouse V1[41] and both acute and longitudinal electrophysiological recordings in cat V1[9,10]. These changes not only reduce FS interneuron firing rates, but also strongly reduce DE-driven cFos expression among PV + and PV- neuron populations

in layers 2/3 (Fig. 3). Thus our data are consistent with the interpretation that MD leads to a transient decrease in cortical inhibition through effects of PV + interneurons[9,41]. Importantly, closure of the critical period for ODP is thought to involve restoration of mature levels of cortical inhibition, which disrupts subsequent competitive plasticity of excitatory inputs[57–60]. We find that while this response depression is almost completely reversed by 5-day BR, only partial recovery of DE responses is achieved with RO. Differential recovery between BR and RO is evident both at the level of

**Fig. 5 Post-BR SD prevents recovery of DE and SE responses after MD. a**, **c** Cumulative distributions of preferred-stimulus DE (**a**) and SE (**c**) visually-evoked firing rate responses for bV1 RS neurons. DE firing rate responses were significantly decreased in BR + SD mice relative to BR + Sleep mice. **b**, **d** Violin plots of RS neurons' DE (**b**) and SE (**d**) visually-evoked responses recorded from neurons in bV1 layers 2/3, 4, or 5/6. Mann–Whitney test, $p = 0.004$, $p = 0.26$, $p = 0.64$ for layers 2/3, 4, and 5/6 in the DE, respectively. Mann–Whitney test, $p = 0.61$, $p = 0.025$, $p = 0.029$ for layers 2/3, 4, and 5/6 in the SE, respectively. **e**, **g** Cumulative distributions of maximal DE (**e**) and SE (**g**) visually-evoked firing rate responses for bV1 FS interneurons. Firing rate responses for DE and SE stimulation were significantly decreased and increased, respectively, in BR + SD mice. **f**, **h** Violin plots of FS interneurons' DE (**f**) and SE (**h**) visually-evoked responses recorded from neurons in bV1 layers 2/3, 4, or 5/6. Mann–Whitney test, $p = 0.66$, $p = 0.032$, $p = 0.83$ for layers 2/3, 4, and 5/6 in the DE, respectively. Mann–Whitney test, $p = 0.24$, $p = 0.49$, $p = 0.029$ for layers 2/3, 4, and 5/6 in the SE, respectively. * and ** indicate $p < 0.05$ and $p < 0.01$, K-S test (**a**, **c**, **e**, and **g**) or Mann–Whitney (**b**, **d**, **f**, and **h**); ns indicates not significant. Values for the MD-only condition (gray dashed lines) from Fig. 2 are shown for comparison. Dashed lines in violin plots (**b**, **d**, **f**, and **h**) represent the 25%, median, and 75% quartiles. Sample sizes per group of units found in figure.

firing rates (Fig. 2) and DE-driven cFos expression (Fig. 3). Across the initial 5-day MD period, both FS interneurons and RS neurons also show widespread potentiation of SE firing rate responses, across all layers of bV1. These SE response changes (which are thought to occur only after DE response depression has already take place)[6,9,10] appear to be almost fully reversed after 5-day BR. In contrast, SE response enhancement is minimally altered after 5-day subsequent RO. In general, these findings are consistent with intrinsic signal imaging studies in binocular mouse V1, which indicated that a single day of BR is superior to RO at restoring binocularity[23].

While the present studies do not include a true functional measure of binocular disparity-driven vision (such as the visual cliff task)[61], we did assess how visual acuity changes for each of the two eyes as a function of MD, BR, and RO. The acuity values we measured in NR mice are virtually identical to those measured previously in juvenile mice using optokinetic responses[52,53]. Consistent with previous findings in mice[53], we find that MD alone leads to selective disturbance of acuity for the DE, while RO (and to a lesser extent, BR as well) further disrupt acuity for the SE. Critically, these findings may be consistent with studies in patients with amblyopia, where fellow eye acuity disruption is common, and recovery of function for the fellow eye is an important consideration[62]. Future studies will be needed to further test the functional outcomes of BR and RO, including additional tests of visual acuity (such as the visual water maze)[54] and use of binocular disparity cues in the context of recovery.

Importantly, for both BR and RO recovery groups, engaged viewing of grating stimuli during their presentation (and sustained wakefulness) was encouraged by placement in an enriched environmental context containing treats, toys, and a running wheel. This strategy was chosen to create a condition that could be feasibly translated to visual therapy for patients (including children) with amblyopia. It is worth noting that both environmental enrichment[63–65], and specifically running wheel access[63,66–68], have been shown to promote ocular dominance plasticity (even well outside of the typical critical period). For example, 2–3 weeks of environmental enrichment in adult rats in conjunction with RO has been shown to restore DE visual acuity and cortical responses[64]. These effects of enrichment are linked to changes in the function of multiple types of V1 interneurons, including those expressing somatostatin (SST) and vasoactive intestinal peptide (VIP)[68], as well as PV + interneurons. Ultimately, the changes we observed in the activation state of PV + interneurons may be mediated by changes in the activity of VIP interneurons, leading to reduced activity of both PV + interneurons and SST + interneurons (i.e., disinhibition). Future studies will be needed to determine: (1) whether environmental enrichment is contributing to recovery of binocular responses in V1 via similar disinhibitory mechanisms, (2) the precise role of FS interneurons' response properties in mediating recovery, and (3) the extent to which initial eye-specific response changes during recovery are also mediated by Hebbian vs. homeostatic plasticity

mechanisms[44,47,48,69–72]. Further, because we see some recovery of V1 binocular responses after both BR and RO, it is plausible that enrichment is beneficial in the context of either type of recovery experience. Future studies will also be needed to test how enrichment effects differ between BR and RO, and how these compare with enrichment effects observed in adult amblyopic rats[64]. Nonetheless, because both BR vs. RO included exposure to identical environmental enrichment over the same time interval, it is clear that enrichment alone is insufficient to facilitate complete recovery over the time frame studied here. Rather, optimal recovery is only observed after enriched *binocular* visual experience in an enriched environment.

How does post-experience sleep or sleep loss affect bV1 during recovery? Our data clearly demonstrate that following periods of BR experience, subsequent sleep is essential for recovery of MD-driven changes in ocular dominance (Fig. 4), DE and SE firing rate responses (Fig. 5), and DE-driven cFos expression (Fig. 6). These data demonstrate that following MD, sleep plays a critical role in restoring normal visual function. Prior work has shown that post-MD sleep is essential for initial ocular dominance shifts in favor of the spared eye[8–11,36] and for MD-induced structural plasticity in V1 neurons[55]. In comparison, the role of sleep in promoting recovery of bV1 function following amblyopia onset is understudied. Prior work done in critical period cats after brief RO indicated that post-RO SD had little impact—and even tended to reduce recovery of binocular vision[37]. However, virtually nothing is known about interactions between BR visual experience and subsequent sleep. While post-BR sleep has been suggested to promote homeostatic downscaling of firing rates in rodent monocular zone[73], no prior work has addressed how it affects bV1 ODP. The present work characterizes how sleep contributes to experience-driven recovery of binocular vision in bV1. We find that post-BR sleep is required for reversal of both DE response depression and SE response potentiation, in both RS neurons and FS interneurons. As with changes driven by initial MD (Fig. 2, Fig. 3), changes driven by post-BR sleep appear to be most dramatic in layers 2/3 (Fig. 5, Fig. 6).

Why might sleep be essential for these changes? Available data suggests that both Hebbian synaptic potentiation and weakening can occur in bV1 during post-MD sleep[9,10,36,74,75] through sleep-dependent activation of specific molecular pathways[10,36,76] or sleep-specific activity patterns[9]. It is plausible that similar mechanisms are involved in the reversal of MD-driven synaptic changes during post-BR sleep. For example, specific oscillatory patterning of neuronal firing in the V1-LGN network during sleep may be essential for spike timing-dependent plasticity between synaptically-connected neurons[34,35,77–80]. Alternatively, sleep may promote permissive changes in biosynthetic pathways that are essential for consolidating some forms of plasticity in vivo[81–83]. In V1, sleep plays a role in increasing inhibition within layers 2/3, reducing E/I ratios across the rest phase[84]; this may play a role in reversing ocular dominance changes driven by suppression of FS interneurons in the

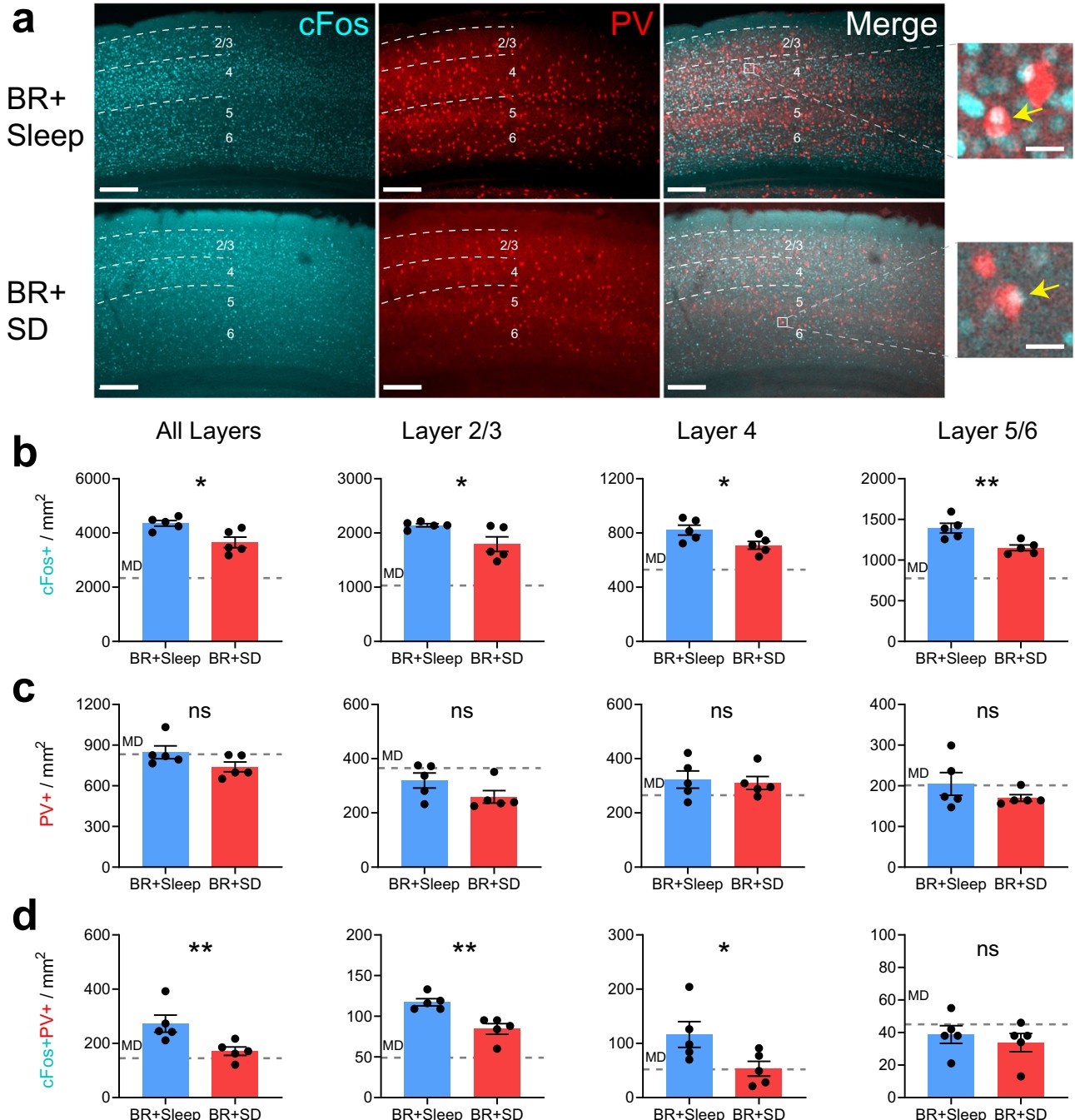

**Fig. 6 Post-BR SD prevents recovery of DE-driven cFos expression in bV1. a** bV1 cFos (cyan) and PV (red) expression after DE stimulation in BR + Sleep and BR + SD mice. Mice (n = 5/treatment group) received DE-only visual stimulation for 30 min, then were returned to their home cages for 90 min prior to perfusion. Scale bar = 100 μm; 20 μm (magnification inset). **b** DE-driven cFos+ neuron density was reduced in BR + SD mice relative to BR + Sleep mice. Unpaired t-test: p = 0.013. cFos+ neuron density was reduced in bV1 layers 2/3, 4, and 5/6 after BR + SD relative to BR + Sleep. Unpaired t-test for layers 2/3, 4, or 5/6, respectively: p = 0.036, p = 0.041, and p = 0.008. **c** PV immunostaining was similar between groups. PV + interneuron density in bV1 layers 2/3, 4, and 5/6 was similar between groups. **d** cFos+ PV + interneuron density was decreased in BR + SD mice relative to BR + Sleep mice. Unpaired t-test: p = 0.020. cFos+ PV + interneuron density was reduced in bV1 layers 2/3 and 4 in BR + SD mice relative to BR + Sleep mice. Unpaired t-test for layers 2/3, 4, or 5/6, respectively: p = 0.003, p = 0.049, and p = 0.53. Values for the MD-only condition (gray dashed lines) from Fig. 3 are shown for comparison. * and ** indicate p < 0.05, and p < 0.01, respectively; ns indicates not significant. Error bars indicate mean ± SEM.

context of MD[9,41]. Finally, sleep also contributes to homeostatic changes in V1 neurons' firing rates[56,85]; thus sleep-dependent homeostatic plasticity may also contribute to bV1 changes observed in BR + Sleep, but not BR + SD, mice. Indeed, spontaneous and mean neuronal firing rate data from these mice (Supplementary Fig. 6) support this idea.

Many factors affect the degree of ODP initiated by MD in animal models of amblyopia, including behavioral state[8,10,11] and neuropharmacology[58,86–88]. Emerging data suggests that these factors may also affect recovery from amblyopia[25,63,87]. However, findings from both patients and animal models have raised debate about whether dominant-eye (SE) patching provides the optimal

sensory stimulus for promoting recovery of vision[28,32,89,90]. Here, in side-by-side comparison of the effects of brief binocular vs. monocular recovery experiences in mice, we show the two have strikingly different effects on bV1 ocular dominance and network activation in bV1. These data reflect findings using comparisons of 24-h binocular vs. monocular recovery[23], where simply re-opening the DE in mice was found to be more efficacious for restoring binocular vision than RO. One possibility is that plasticity mechanisms in mouse bV1 differ from those in operation in primate bV1 when the DE is reopened[17]. In support of this argument, simply reversing occlusion of the weaker eye is insufficient to correct amblyopia in patients. On the other hand, our present data suggest that features of the binocular visual stimulation used in our BR condition (including presentation of visual stimuli simultaneously to the two eyes, in the context of engaging and arousing environmental stimulation) may be useful for developing experimental therapeutics for amblyopia recovery. Critically, our data also demonstrate that the timing of sleep relative to visual experience during amblyopia treatment may be an important consideration for restoring normal bV1 function. This finding may have important implications for treating amblyopia later in life, after the critical period, and future studies should address whether similar or different mechanisms are in operation when recovery occurs long after the critical period has closed. While the use of brief critical period MD in animal models may have some limitations (including the brevity of visual occlusion compared to visual disruptions such as congenital cataract in infancy)[91], we hope that our present data will inform future strategies for optimizing amblyopia treatment in children.

## Methods

**Animal housing and husbandry.** All mouse husbandry and experimental/surgical procedures were reviewed and approved by the University of Michigan Internal Animal Care and Use Committee. Weaned C57BL6/J mice were housed in a vivarium under 12 h:12 h light/dark cycles (lights on at 9AM) with littermates starting postnatal day 21 and had *ad lib* access to food and water. After eyelid suture surgeries for monocular deprivation (MD) at postnatal day 28 (P28), mice were single housed in standard cages with beneficial environmental enrichment. For studies comparing the effects of sleep on binocular recovery (BR) visual experience, mice were housed with a 4 h:20 h light:dark cycle (lights on from 9AM-1PM during visual enrichment, dim red light outside of visual enrichment) and had *ad lib* access to food and water during this period.

**Monocular deprivation, recovery, visual enrichment, and sleep deprivation.** For all experiments, male littermates were randomly assigned to treatment groups. For monocular deprivation (MD), mice were anesthetized at P28 using 1–1.5% isoflurane. Nylon non-absorbable sutures (Henry Schein) were used to occlude the left eye. Sutures were checked twice daily to verify continuous MD; during this time they were handled 5 min/day. After MD (at P33), mice were anesthetized with 1–1.5% isoflurane a second time and left eyelid sutures were removed. Mice that underwent binocular recovery (BR) were then housed over the next 5 days with both eyes open; during this time, they were handled daily for 5 min/day. Mice that underwent reverse occlusion (RO) had the right (previously spared; SE) eye sutured for the next 5 days; these mice were also handled 5 min/day during this period. Mice that lost sutures during the MD or recovery periods or developed eye abnormalities were excluded from the study. BR and RO mice underwent a 5-day period of identical daily enriched visual experience from P34-38. This regimen consisted of a daily placement in a 15" × 15" × 15" Plexiglas arena surrounded by 4 high-contrast LED monitors, from ZT0 (lights on) to ZT4. Phase-reversing oriented grating stimuli (100% contrast, 1 Hz reversal frequency) of 8 orientations were presented repeatedly on the 4 monitors in a random, interleaved fashion. Phase-reversing gratings were chosen over drifting gratings because: (1) phase-reversing gratings have been shown to induce rapid stimulus-driven plasticity in adult mouse V1[77,92–96], (2) phase-reversing gratings are superior at driving high-amplitude, synchronous activation of visual inputs to V1[97,98], and (3) phase-reversing gratings were more straightforward to present to mice in a coordinated manner within the square arena. Spatial frequencies for grating stimuli varied from 0.0025-0.1 cycles/deg during this period, depending on the mouse's position within the arena. During this 4-h period of daily visual enrichment, mice were encouraged to voluntarily remain awake, explore the chamber, and view visual stimuli via presentation of a variety of enrichment items (novel objects, transparent tubes, and a flying saucer-style running wheel, to which mice had *ad lib* access) and palatable

treats. This experimental strategy was aimed at maximizing time spent voluntarily awake, while minimizing experimenter mechanical interventions[99]. For sleep deprivation (SD) studies on BR experience, immediately following the 4-h visual enrichment period, mice were placed in their home cage within a sound-attenuated behavioral chamber (Med Associates) under dim red light (Fig. 4b). BR + Sleep and BR + SD mice were housed under the same conditions, with BR + SD undergoing SD by gentle handling for the first 4 h post-enrichment[77,85]. Ambient red light levels (measured at 530-980 nm) during this period were sufficiently low ($\leq 3.68 \times 10^9$ photons/cm$^2$/s) that mice would have negligible additional visual experience (i.e., form vision) during SD, based on published psychometric data[100]. Briefly, gentle handling procedures involved visually monitoring the mice for assumption of sleep posture—i.e., huddled in their nest with closed eyes. Upon detection of sleep posture, the cage was either tapped or (if necessary) shaken briefly (1–2 s). If sleep posture was maintained after these interventions, the nesting material within the cage would be moved using a cotton-tipped applicator. No novel objects or additional sensory stimuli were provided, to limit sensory-based neocortical plasticity during sleep deprivation procedures[101]. To estimate the amount of sleep lost in BR + SD mice, BR + Sleep mice were visually monitored every 5 min over the first 4 h following visual enrichment for assumption of sleep postures, similar to previous studies[83,102–104]. As previously described, similar procedures used for sleep deprivation in adult mice[105] and critical period cats[8] either have no significant effect on serum cortisol, or increases it to a degree that is orders of magnitude lower than that capable of disrupting ODP[106].

**In vivo neurophysiology and single unit analysis.** Mice underwent stereotaxic, anesthetized recordings using a combination of 0.5-1.0% isoflurane and 1 mg/kg chlorprothixene (Sigma). A small craniotomy (1 mm in diameter) was made over right-hemisphere bV1 (i.e., contralateral to the original DE) using stereotaxic coordinates 2.6–3.0 mm lateral to lambda. Recordings of neuronal firing responses were made using a 2-shank, linear silicon probe spanning the layers of bV1 (250 μm spacing between shanks, 32 electrodes/shank, 25 μm inter-electrode spacing; Cambridge Neurotech). The probe was slowly advanced into bV1 until stable, consistent spike waveforms were observed on multiple electrodes. Neural data acquisition using a 64-channel Omniplex recording system (Plexon) was carried out for individual mice across presentation of visual stimuli to each of the eyes, via a full field, high-contrast LED monitor positioned directly in front of the mouse (i.e., within the binocular visual field). Recordings were made for the right and left eyes during randomly interleaved presentation of a series of phase-reversing oriented gratings (8 orientations + blank screen for quantifying spontaneous firing rates, reversing at 1 Hz, 0.05 cycles/degree, 100% contrast, 10 s/stimulus; Matlab Psychtoolbox). Spike data for individual neurons was discriminated offline using previously described PCA and MANOVA analysis[77,85,92,107] using Offline Sorter software (Plexon). Following artifact removal, clusters of spike waveforms were discriminated based on waveform shape and amplitude, position in three-dimensional PCA space, and apparent neuronal subclass (i.e., FS interneuron vs. principal [regular spiking] neuron). Cluster separation was verified using MANOVA on the first three principal component values ($p < 0.05$). Subclass discrimination was based upon spike waveform half-width, which was narrower for FS interneurons than principal neurons, and neuronal firing rate, which was higher for FS interneurons than principal neurons. Using both parameters together revealed distinct clusters corresponding to the two cell types (Supplementary Fig. 1 and 5), consistent with previous reports[108–110]. Spike sorting and subsequent analysis of firing response properties was carried out by a scorer blinded to each animal's experimental group. Only those neurons which maintained stable spike waveforms, with consistent firing throughout presentation of stimuli to both eyes, were included in subsequent analysis. Because the inter-electrode distance on the silicon probes could allow the same neuron's spike waveforms to be detected on more than one electrode, spike trains recorded for units on adjacent electrodes which had identical spike timing were omitted from subsequent analyses.

For each neuron, a number of response parameters were calculated[9,10]. Firing rate-based comparisons were made using firing rate responses recorded for oriented grating stimulus (or blank screen presentation), for each eye, averaged across all presentations (i.e., 8 presentations for 10 s each; 80 s total). For analysis of DE and SE maximal firing rates (Fig. 2), values were compared for mean firing rate across the 80 s of presentation for each neuron's preferred stimulus orientation. Comparisons of mean and spontaneous firing rates (Supplementary Figs. 2 and 6) were made on neuronal data averaged across all presentations of all oriented grating stimuli (i.e., preferred and non-preferred), or all blank screen presentations, respectively, for each of the two eyes. Visual responsiveness was assessed by comparing each neuron's spontaneous firing rates during blank screen presentations with evoked firing during grating presentations of the preferred orientation. Neurons with spontaneous firing higher than maximum evoked firing were considered non-visually responsive. An ocular dominance index was calculated for each visually responsive unit as $(C-I)/(C+I)$ where C represents the maximal visually-evoked firing rate for preferred-orientation stimuli presented to the contralateral (left/deprived) eye and I represents the maximal firing rate for stimuli presented to the ipsilateral (right/spared) eye. Ocular dominance index values range from −1 to +1, where negative values indicate an ipsilateral (SE) bias, positive values indicate a contralateral (DE) bias, and values close to 0 indicate

similar responses for stimuli presented to either eye. For additional visual comparisons of ocular dominance distributions, neuronal responses were categorized on a 7-point OD scale, as follows: $-1$ to $-0.75 = 7$; $-0.75$ to $-0.45 = 6$; $-0.45$ to $-0.15 = 5$; $-0.15$ to $0.15 = 4$; $0.15$ to $0.45 = 3$; $0.45$ to $0.75 = 2$; and $0.75$ to $1 = 1$. Finally, for each mouse, a contralateral bias index (CBI) was calculated based on all neuronal responses recorded, as: CBI $= [(n_1 - n_7) + 2/3(n_2 - n_6) + 1/3(n_3 - n_5) + N] / 2N$, where $N =$ the total number of neurons recorded, and $n_x =$ the number of neurons with a given OD score $x$ on the 7-point scale.

**Optokinetic behavioral assay.** Visual acuity was measured for the right and left eyes by measuring optokinetic responses using an OptoMotry tracking system (Cerebral Mechanics, Inc.)[52]. For optokinetic measures, mice were placed on an elevated platform in the center of an enclosed arena surrounded by four LED monitors displaying a clockwise or counterclockwise drifting vertical sine wave gratings. Gratings were presented at multiple spatial frequencies at 100% contrast, in a randomly interleaved manner, and visual tracking behavior was measured by an expert scorer blind to each mouse's experimental condition. Acuity was measured for each of the two eyes based on the spatial frequency threshold at which clockwise or counterclockwise tracking behavior ceased. Acuity values measured for NR mice using this method (Supplementary Fig. 4) were similar to those reported previously for critical period mice[52].

**Histology and immunohistochemistry.** Following all electrophysiological recordings, mice were euthanized and perfused with ice cold PBS and 4% paraformaldehyde. Brains were dissected, post-fixed, cryoprotected in 30% sucrose solution, and frozen for sectioning. 50 µm coronal sections containing bV1 were stained with DAPI (Fluoromount-G with DAPI; Southern Biotech). DAPI staining provided contrast for identifying sites of electrode shank penetration into the tissue, and for approximating stereotaxic coordinates of shank locations to verify placement within bV1 (Figs. 1b and 4c). Mice whose electrode placement could not be verified were excluded from further analyses.

For immunohistochemical quantification of PV and DE-driven cFos expression in bV1, mice from all groups underwent monocular eyelid suture of the original SE (i.e., the right eye) at ZT12 (lights off) the evening before visual stimulation. At ZT0 (next day), stimulation of the original DE (i.e., the left eye) was carried out in the LED-monitor-surrounded arena with treats and toys to maintain a high level of arousal (as described above) Mice from all groups were exposed to a 30-min period of oriented gratings (as described for visual enrichment above), after which they were returned to their home cage for 90 min (for maximal visually-driven cFos expression) prior to perfusion. Coronal sections of bV1 were collected as described above and immunostained using rabbit-anti-cFos (1:1000; Abcam, ab190289) and mouse-anti-PV (1:2000; Millipore, MAB1572) followed by secondary antibodies: Alexa Fluor 488 (1:200; Invitrogen, A11032) and Alexa Fluor 594 (1:200; Invitrogen, A11034). Stained sections were mounted using Prolong Gold antifade reagent (Invitrogen) and imaged using a Leica SP8 confocal microscope with a 10× objective, to obtain z-stack images (10 µm steps) for maximum projection of fluorescence signals. Identical image acquisition settings (e.g. exposure times, frame average, pixel size) were used for all sections. bV1 boundaries were estimated using comparisons to established stereotaxic coordinates. cFos+ and PV + cell bodies were quantified in 3-4 sections (spanning the anterior-posterior extent of bV1) per mouse by a scorer blinded to the animal's experimental condition, and reported as approximate density. Co-labeling was quantified using the Image J JACoP plugin[111] and values for each mouse are averaged across 3-4 sections. As an added control (to ensure that observed changes in expression in bV1 were the result of visual manipulations; Figs. 3 and 6), primary auditory cortex (A1) was identified in the same brain sections used for bV1 measurement. A1 boundaries were estimated using established stereotaxic coordinates. A1 layer-specific expression was quantified in each section as described above (Supplementary Figs. 3 and 7).

**Statistics and reproducibility.** Statistical analyses were carried out using GraphPad Prism software (Version 9.5). Comparisons of ocular dominance metrics, firing rates, and visual response properties were made using stably recorded (i.e., with consistent waveforms present across the entire visual stimulation period, visually responsive (visually-evoked firing rate >spontaneous firing rate) units in bV1. Each neuron recorded was considered an independent sample for analysis of ocular dominance and firing rate data; mean values were also calculated on a per-animal basis as appropriate. Two-tailed, unpaired $t$-tests were used to assess the difference between two groups. One-way ANOVAs with Tukey's *post hoc* test for pairwise comparison or comparisons of selected groups was used for comparisons across more than two treatment groups. Nonparametric tests such as Kruskal–Wallis with Dunn's *post hoc* test and Mann–Whitney test were used as appropriate, for non-normal data distributions. Cumulative distribution data was assessed using the Kolmogorov–Smirnov (K–S) test. Specific statistical tests for each data set and p-values can be found within the corresponding figure legends.

**Reporting summary.** Further information on research design is available in the Nature Portfolio Reporting Summary linked to this article.

## Data availability
Source data underlying main figures are provided in Supplementary Data 1. All other relevant raw data is available upon reasonable request from the corresponding author.

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

## Acknowledgements

We thank Abbey Roelofs (LSA Information and Technology Services) for software programming assistance. Graphics in Figs. 1a, b and 4a, b and Supplementary Fig. 4a were created with Biorender.com. We acknowledge support from the Kellogg Eye Center Vision Research Core funded by P30 EY007003 from the National Eye Institute for optokinetic behavioral assay assistance. This work was supported by a University of Michigan Rackham Graduate School Candidate Grant and Merit Fellowship to J.D.M., NIH R01NS104776, and a Research to Prevent Blindness Walt and Lilly Disney Award for Amblyopia Research to S.J.A.

## Author contributions

J.D.M. and S.J.A. designed the experiments. J.D.M. carried out electrophysiological experiments with assistance from M.J.D., D.S.P., B.C.C., and S.J.A. J.D.M. carried out immunohistochemical experiments with assistance from D.T., L.G.W., W.P.B., and S.J. J.D.M., S.S., and C.L. carried out optokinetic behavior experiments. J.D.M. analyzed the data. J.D.M. and S.J.A. wrote the manuscript.

## Competing interests

The authors declare no competing interests.
