## [Peer Review File · Communications Biology]

Reviewers' comments:

Reviewer #1 (Remarks to the Author):

In this very interesting study, the authors show convincingly in mice that the deleterious effects of brief MD are reversed better by restoring binocular visual experience than by reverse occlusion. An important and novel feature of the experimental design is careful control of visual experience in the recovery period, which is designed to promote synchronous activation of the two eyes. Further, the authors show that the recovery is facilitated by sleep, and impaired by sleep deprivation. The data are of extremely high quality, yielding insights into how responses change in both regular spiking principal neurons and in fast spiking interneurons, across all layers of bV1. The main findings are nicely corroborated by cFos staining. In my opinion, this paper sets the standard for the field of ocular dominance plasticity in mice. I found the discussion to be balanced and appropriately circumspect about the relevance to human amblyopia. One small comment is that it would be worthwhile to acknowledge that brief MD, from P28 to 33, may not actually be an adequate model for amblyopia pathogenesis, which typically arises from birth in humans. It remains to be determined the extent to which these findings would apply to ocular dominance changes elicited by many weeks of MD, initiated at the time of eye opening in mice. However, this is a small point. Overall, the authors are to be congratulated on producing a very nice paper that substantially advances the field.

Reviewer #2 (Remarks to the Author):

In their manuscript the authors report that, in a mouse model of amblyopia, binocular sight conditions lead to a better recovery of visual functions with respect to reverse occlusion. Moreover, they authors report that sleep during a period after visual stimulation is necessary to obtain visual function recovery.

The present work is of potential interest for the field, since amblyopia is one of the major causes of visual defects in humans. Currently, studies in experimental amblyopia are increasingly focused on the relative role of monocular vs. binocular conditions as suitable strategies for application to patients, and also the role of sleep on ocular dominance plasticity is certainly relevant. The authors applied well-established techniques to investigate visual cortical plasticity, such as electrophysiological single unit recordings and HIC staining for c-FOS. But at the moment, there are some crucial issues regarding to what extent the proposed hypotheses have been demonstrated, the figure quality and explanation of results, that need to be improved, together with the need to perform further experiments, in order to clarify essential points (see below). Moreover, the present work lacks essential control groups.

Major

- The visual enrichment protocol used and described in the M&M section (visual+wheels+toys) is actually quite confusing. It seems an attempt to obtain precisely controlled visual enrichment (which is good), but with many confounding factors, such as the presence of toys and of running wheels. This does not allow to understand the role of these different components in the recovery effects, as running by itself is a strong promoter of visual cortex plasticity. The authors need to separate these environmental components, by adding additional groups in which one single variable (e.g., physical activity vs. visual enrichment) is addressed independently at each time.

- Also related to the previous point. Physical exercise recently emerged as an effective treatment for amblyopia in rodents. Which was the amount of daily motor activity performed on the running wheels in the present work?

- While the authors include a group of naïve animals as controls, additional essential controls are needed. In particular, the authors must include groups of either binocular vs. reverse occlusion conditions of recovery in the absence of enrichment. In these additional groups, the authors also need to ask the specific role of sleep, performing analysis of visual recovery in standard reared groups (no enrichment) in which sleep deprivation is applied.

- The manuscript does not report any effect on visual perceptual abilities. Amblyopia results in

major deficits concerning visual acuity and visual depth perception. The authors should also measure these parameters, as the present results, albeit of interest, do not allow to understand the real impact of visual enrichment and sleep on visual function recovery.

- The authors report that “-During this 4-h period of daily visual enrichment, mice were encouraged to remain awake...”. It is presently totally unclear how this encouragement was performed.

- The authors only focused on PV+ interneurons. What about the expression and activity of other subclasses of GABAergic interneurons, known to play key roles in visual cortex plasticity (e.g., VIP+ cells)?

- Methods. The contralateral bias index (CBI) is not mentioned in M&M, nor introduced in the results. It only appears in figures and legends. The authors need to expand all sections regarding this essential parameter.

- Methods. It is presently unclear which method was followed by the authors to distinguish between regular spiking and fast spiking cells. Presently, in the M&M section the authors refer to a previous reference, but this is not enough.

Minor

- The presence in the text of a huge number of abbreviations renders the manuscript hard to read; I suggest reducing the number of abbreviations adding a table with a complete list of them.

- 25µm of inter-electrodes distance is not enough to guarantee that the same cell is not recorded by different contact points; how the authors handled this fact?

- I am puzzled by the choice of phase reversal gratings for spike analysis: phase reversal stimuli are usually utilized for field potentials, while drifting gratings are mainly utilized in spike analysis since they likely engage a higher number of receptive fields (M Self 2014; Niell and Stryker 2008; Niell and Stryker 2010; Herikstad 2011; Mazziotti 2017; Sun 2021; Matteucci 2021). Why did the authors decide on a different approach?

- The paper lacks important quotations of recent work concerning binocular recovery and visual enrichment. For instance, in the following sentence: “More recently, studies in developing cats and rodents have found that under certain conditions the role of binocular vision in restoring binocularity of responses in bV1”, please quote recent work by Sansevero et al. The same applies to this sentence “Insofar as MD serves as a model for amblyopia caused by disruption of vision in one of the two eyes during childhood, these data add to a body of growing evidence that suggests that enriched binocular visual experience may offer advantages over and above the standard of care for amblyopia”.

Response to reviewers' comments:

Reviewer #1 (Remarks to the Author):

In this very interesting study, the authors show convincingly in mice that the deleterious effects of brief MD are reversed better by restoring binocular visual experience than by reverse occlusion. An important and novel feature of the experimental design is careful control of visual experience in the recovery period, which is designed to promote synchronous activation of the two eyes. Further, the authors show that the recovery is facilitated by sleep, and impaired by sleep deprivation. The data are of extremely high quality, yielding insights into how responses change in both regular spiking principal neurons and in fast spiking interneurons, across all layers of bV1. The main findings are nicely corroborated by cFos staining. In my opinion, this paper sets the standard for the field of ocular dominance plasticity in mice. I found the discussion to be balanced and appropriately circumspect about the relevance to human amblyopia.

We are very grateful to the reviewer for their positive comments on our work.

One small comment is that it would be worthwhile to acknowledge that brief MD, from P28 to 33, may not actually be an adequate model for amblyopia pathogenesis, which typically arises from birth in humans. It remains to be determined the extent to which these findings would apply to ocular dominance changes elicited by many weeks of MD, initiated at the time of eye opening in mice. However, this is a small point. Overall, the authors are to be congratulated on producing a very nice paper that substantially advances the field.

We agree completely that brief MD as a model for long-lasting visual disruption leading to amblyopia in children has limitations that could impact its external validity. Because this is a point worth making clearly, we have modified the final paragraph of the Discussion section to do so. We appreciate this helpful feedback.

Reviewer #2 (Remarks to the Author):

In their manuscript the authors report that, in a mouse model of amblyopia, binocular sight conditions lead to a better recovery of visual functions with respect to reverse occlusion. Moreover, they authors report that sleep during a period after visual stimulation is necessary to obtain visual function recovery. The present work is of potential interest for the field, since amblyopia is one of the major causes of visual defects in humans. Currently, studies in experimental amblyopia are increasingly focused on the relative role of monocular vs. binocular conditions as suitable strategies for application to patients, and also the role of sleep on ocular dominance plasticity is certainly relevant. The authors applied well-established techniques to investigate visual cortical plasticity, such as electrophysiological single unit recordings and HIC staining for c-FOS. But at the moment, there are some crucial issues regarding to what extent the proposed hypotheses have been demonstrated, the figure quality and explanation of results, that need to be improved, together with the need to perform further experiments, in order to clarify essential points (see below). Moreover, the present work lacks essential control groups.

Major

- The visual enrichment protocol used and described in the M&M section (visual+wheels+toys) is actually quite confusing. It seems an attempt to obtain precisely controlled visual enrichment (which is good), but

with many confounding factors, such as the presence of toys and of running wheels. This does not allow to understand the role of these different components in the recovery effects, as running by itself is a strong promoter of visual cortex plasticity. The authors need to separate these environmental components, by adding additional groups in which one single variable (e.g., physical activity vs. visual enrichment) is addressed independently at each time.

We appreciate this comment and apologize for providing insufficient discussion of rationale for our chosen technique. Our overall strategy was aimed at maximizing time spent awake (and thus receiving visual input from the surrounding LED monitors) during the first few hours of the light phase, without direct (i.e. mechanical) interventions to disrupt sleep. We also hoped to increase the translational potential with respect to the strategy for maintaining interaction with the visual stimuli (i.e., one would aim to motivate a child to use their eyes in this way, rather than keeping them engaged through mechanical interventions like tapping or shaking them!). We have made this clearer in the manuscript where we discuss the methodology.

We completely agree with the reviewer's comment that physical activity and environmental enrichment can both contribute to sensory plasticity, possibly in a manner independent of the specific visual stimuli presented. Assessing aspects of recovery driven by these aspects vs. the stimuli themselves is an interesting and important issue, but far beyond the scope of this study. Here, we were focused on differences between binocular vs. monocular presentation, and differences due to post-experience sleep, independent of other aspects of recovery. For this reason, we provided the very same enrichment to animals in all recovery groups. Critically, and as we now discuss in the revised text, the fact that we do not see full recovery in mice with monocular vision, and in sleep deprived mice, suggests that environmental enrichment and exercise alone are inadequate for optimizing recovery.

We appreciate the reviewer pointing out that we did not adequately discuss how other aspects of our recovery condition, such as access to a running wheel, could contribute to recovery overall. We now provide discussion of these issues and how they might relate to the translation of our findings.

- Also related to the previous point. Physical exercise recently emerged as an effective treatment for amblyopia in rodents. Which was the amount of daily motor activity performed on the running wheels in the present work?

Mice had *ad lib* access to a "flying saucer" style wheel during visual stimulus presentation periods. As described in response to the previous point, *ad lib* access was used to maximize awake (voluntary) interaction with visual stimuli, rather than as an experimental variable in these studies.

- While the authors include a group of naïve animals as controls, additional essential controls are needed. In particular, the authors must include groups of either binocular vs. reverse occlusion conditions of recovery in the absence of enrichment. In these additional groups, the authors also need to ask the specific role of sleep, performing analysis of visual recovery in standard reared groups (no enrichment) in which sleep deprivation is applied.

As described above, clarifying the differential roles of enrichment vs. the eye-specificity of visual stimuli (and the role of subsequent sleep) in regulating V1 function overall (what the reviewer alludes to here)

is well beyond the scope of any single manuscript. For this reason, where enrichment could have effects with respect to recovery (i.e., in recovery mice), we kept this feature constant.

- The manuscript does not report any effect on visual perceptual abilities. Amblyopia results in major deficits concerning visual acuity and visual depth perception. The authors should also measure these parameters, as the present results, albeit of interest, do not allow to understand the real impact of visual enrichment and sleep on visual function recovery.

We completely agree that in patients with amblyopia, the major clinical targets are acuity and depth perception using binocular disparity cues. In response to this suggestion, we have tested visual acuity for the spared and deprived eyes in our NR, MD, BR, and RO mice, to compare effects of initial MD and recovery experiences on acuity. We find that, consistent with prior findings, MD leads to reduced deprived eye acuity. This reduction in acuity is maintained in RO, but not BR, mice. Also consistent with prior findings, we find that RO leads to a massive reduction in spared eye acuity, which is much less extreme in BR mice. Taken together, this suggests that BR is superior to RO at preventing degradation of acuity after MD.

- The authors report that “-During this 4-h period of daily visual enrichment, mice were encouraged to remain awake...”. It is presently totally unclear how this encouragement was performed.

We thank the reviewer for pointing out that this was unclear as written – this is helpful feedback! We hope that our response to the first point above clarifies this issue. Mice were encouraged to remain awake based on their voluntary interaction with running wheels and exploration of novel objects in the arena where visual stimuli were presented. We have clarified this in the text, both in the Methods section and elsewhere.

- The authors only focused on PV+ interneurons. What about the expression and activity of other subclasses of GABAergic interneurons, known to play key roles in visual cortex plasticity (e.g., VIP+ cells)?

Here, we focused on the role of PV+ interneurons for two reasons – first, they are known to correspond to FS interneurons that we can identify electrophysiologically, second, they are known to gate ODP. We appreciate the reviewer pointing out that other interneuron types (such as VIP+ interneurons) are involved in regulating ODP, and agree that discussion of this issue is important for contextualizing our present data. We have added substantially to the Discussion section of the manuscript to provide more details about what is known about how these other interneuron types contribute to ODP, and how those functions could relate to our present findings.

- Methods. The contralateral bias index (CBI) is not mentioned in M&M, nor introduced in the results. It only appears in figures and legends. The authors need to expand all sections regarding this essential parameter.

We are grateful to the reviewer for pointing out this glaring omission on our part! We now include full details on the calculation of CBI based on the OD distribution of neurons recorded from each animal.

- Methods. It is presently unclear which method was followed by the authors to distinguish between regular spiking and fast spiking cells. Presently, in the M&M section the authors refer to a previous reference, but this is not enough.

We appreciate the reviewer pointing out that as written, our methodology for spike cluster sorting and cell type classification was unclear. We have added more detailed information to the Methods section and two new supplemental figures to address this. Briefly, FS interneurons were discriminated from principal neurons based on their (narrower) spike waveform and (higher) firing rate – we now show how FS interneurons can be identified using a combination of these metrics (Supplemental Figures 1 and 5). We are grateful for this helpful feedback.

Minor

- The presence in the text of a huge number of abbreviations renders the manuscript hard to read; I suggest reducing the number of abbreviations adding a table with a complete list of them.

We thank the reviewer for raising this concern, which we agree could adversely affect the readability of the manuscript. We have added a table of abbreviations to the front of the manuscript for easier reference.

- 25 μ m of inter-electrodes distance is not enough to guarantee that the same cell is not recorded by different contact points; how the authors handled this fact?

The reviewer is quite correct – with an electrode spacing of 25 μ m, it is plausible that the same neuron's spike data could be detected on adjacent electrodes. We attempted to filter such recordings from data sets included from our analyses by removing units recorded from adjacent electrodes with identical spike timings. We have added this information to the revised Methods section, and thank the reviewer for pointing out this important issue.

- I am puzzled by the choice of phase reversal gratings for spike analysis: phase reversal stimuli are usually utilized for field potentials, while drifting gratings are mainly utilized in spike analysis since they likely engage a higher number of receptive fields (M Self 2014; Niell and Stryker 2008; Niell and Stryker 2010; Herikstad 2011; Mazziotti 2017; Sun 2021; Matteucci 2021). Why did the authors decide on a different approach?

This is an excellent point, and we agree that the use of drifting gratings is optimal for generating higher firing rates in a larger proportion of neurons over time. We had multiple reasons why phase-reversing gratings were more optimal for our experiment. First, phase-reversing gratings were more straightforward to present in a coordinated manner due to the nature (and geometry) of our apparatus for presentation of visual stimuli in a freely-moving condition. Second, phase-reversing gratings are more optimal for synchronous (i.e., phase-locked) activation of V1 neurons. Higher synchrony of retinal and thalamic (i.e., V1 input) activity during presentation of contrast-reversing vs. drifting gratings has been reported previously, and we now cite this in the revised Materials and Methods section. Finally, phase-reversing gratings have been used successfully by our lab and others to drive rapid response plasticity in adult mouse V1.

- The paper lacks important quotations of recent work concerning binocular recovery and visual enrichment. For instance, in the following sentence: "More recently, studies in developing cats and rodents have found that under certain conditions the role of binocular vision in restoring binocularity of responses in bV1", please quote recent work by Sansevero et al. The same applies to this sentence "Insofar as MD serves as a model for amblyopia caused by disruption of vision in one of the two eyes during childhood, these data add to a body of growing evidence that suggests that enriched binocular visual experience may offer advantages over and above the standard of care for amblyopia".

This is an excellent suggestion, and we have modified our references accordingly. We thank the reviewer for pointing this out to us.

Reviewers' comments:

Reviewer #2 (Remarks to the Author):

I appreciate the author willingness in replying to my previous concerns. The manuscript has improved.

There are still two points that need further clarification:

- The authors now measured visual acuity using an optokinetic task that, however, is likely to best portrays activity from subcortical centers, which is not good when working in the amblyopia domain. Accordingly, the reported visual acuity values are quite low even for naïve animals. This limit should be well discussed; alternatively, an operant task like the visual water box task should be employed.

- The authors state (lines 604 and following ones) that "Nonetheless, because both BR vs. RO included exposure to identical environmental enrichment, it is clear that enrichment alone is insufficient to complete recovery over the time frame studied here. Rather, optimal recovery is only observed after enriched binocular visual experience in an enriched environment". How do the authors read their data with respect to the work of Sale et al. (2007, Nature Neuroscience), in which a full recovery was shown in adult amblyopic rats (i.e., a species even less plastic than mice in adulthood) exposed with enrichment while subjected to reverse-suture? Is the difference between the two studies possibly due to different amounts of exposure to enriched stimuli? Indeed, in the present paper enrichment was applied for 4 hrs per day, while in the Sale et al. work it was continuous. If so, please rephrase the Discussion in order to better explain when enrichment might be successful, even in reverse-suture conditions, and when not.

Reviewers' comments:

Reviewer #2 (Remarks to the Author):

I appreciate the author willingness in replying to my previous concerns. The manuscript has improved.

We appreciate the reviewer's previous comments, and are pleased that we were able to use them to improve the manuscript. We thank them for providing us with additional helpful suggestions, which we have addressed below.

There are still two points that need further clarification:

- The authors now measured visual acuity using an optokinetic task that, however, is likely to best portrays activity from subcortical centers, which is not good when working in the amblyopia domain. Accordingly, the reported visual acuity values are quite low even for naïve animals. This limit should be well discussed; alternatively, an operant task like the visual water box task should be employed.

The reviewer makes an excellent point, in that the optometry task is not measuring the use of binocular vision, which is most applicable to full recovery from occlusion-mediated amblyopia. The use of binocular disparity cues to drive behavior has been measured in mice using tasks such as visual cliff, which we do not use here.

We also agree the acuity values we report here for critical period mice, while virtually identical to those reported elsewhere (compare data from NR mice with those shown in Prusky et al. 2004), are lower than those typically reported for rats using similar methodology (e.g., Deidda et al., 2015). These values are also slightly lower, albeit still very similar, when compared against visual acuity measured using the visual water maze task with adult mice (Hosang et al., 2018; Prusky et al., 2000). We believe that the former may reflect true differences in acuity visual between rats and mice (Prusky et al., 2000). On the other hand, the latter may reflect either changes in acuity with brain maturation, or non-sensory aspects of the task (e.g. attention) that make it more difficult for juvenile mice to perform the OptoMotry task. However, it is important to note that Prusky et al found nearly identical acuity measures for adult mice on visual water maze and OptoMotry tasks (Prusky et al., 2000, 2004).

We discuss both of these issues in more detail in the revised Methods, Results, and Discussion sections, and describe future directions using a true binocular vision-based task to assess the functional consequences of the manipulations used here.

- The authors state (lines 604 and following ones) that “Nonetheless, because both BR vs. RO included exposure to identical environmental enrichment, it is clear that enrichment alone is insufficient to complete recovery over the time frame studied here. Rather, optimal recovery is only observed after enriched binocular visual experience in an enriched environment”. How do the authors read their data with respect to the work of Sale et al. (2007, Nature Neuroscience), in which a full recovery was shown in adult amblyopic rats (i.e., a species even less plastic than mice in adulthood) exposed with enrichment while subjected to reverse-suture? Is the difference between the two studies possibly due to different amounts of exposure to enriched stimuli?

Indeed, in the present paper enrichment was applied for 4 hrs per day, while in the Sale et al. work it was continuous. If so, please rephrase the Discussion in order to better explain when enrichment might be successful, even in reverse-suture conditions, and when not.

We appreciate the reviewer raising this point, and we agree that the findings of the Sale et al. study show that with 2-3 weeks of enrichment, there is a benefit for recovery in the case of reverse-suture (i.e., RO). While we cannot assume the mechanisms are identical between the two cases (in adult rats with RO, over a longer window of time, vs. critical period mice with brief BR vs. RO), it is clear that enrichment is improving recovery of binocular function in both cases. We now state this more clearly in the section of our revised Discussion related to enrichment effects. We also make it more clear that enrichment could also be benefitting our RO mice, which do show partial recovery across the 5 d of recovery experience. We thank the reviewer for this comment, which we have used to make a more balanced discussion of this topic.